# Psychological Experience of Smoking Addiction in Family and Friends of Schizophrenic Adults Who Smoke Daily: A Qualitative Study

**DOI:** 10.3390/healthcare11050644

**Published:** 2023-02-22

**Authors:** Pasquale Caponnetto, Marilena Maglia, Annalisa Gulizia, Graziella Chiara Prezzavento, Riccardo Polosa, Maria Catena Quattropani, Maria Salvina Signorelli

**Affiliations:** 1Department of Educational Sciences, University of Catania, 95121 Catania, Italy; 2Center of Excellence for the Acceleration of Harm Reduction (COEHAR), University of Catania, 95121 Catania, Italy; 3Department of Clinical and Experimental Medicine, University of Catania, 95121 Catania, Italy

**Keywords:** schizophrenia, smoking addiction, e-cigarette, thematic analysis, qualitative research

## Abstract

The smoking addiction of patients with severe mental disorders has consequences not only for the patients but also for the people around them. This is qualitative research on family and friends of patients with Schizophrenia spectrum disorders to investigate their perception and vision of smoking, its impact on the patients’ physical and mental health, and the possible attempts to combat addiction. The research also investigates the participants’ views on electronic cigarettes as a means of replacing traditional cigarettes and helping the patient to quit smoking. The survey method used was a semi-structured interview. The answers were recorded, transcribed and analyzed with the technique of thematic analysis. The results of this study show that the view of most participants on smoking is negative (83.3%), although not all of them consider smoking cessation treatments for these patients of primary importance (33.3%). Nevertheless, a good number of them have tried to intervene spontaneously with their own resources and strategies (66.6%). Finally, low-risk products, and in particular electronic cigarettes, are considered by many participants as a useful alternative to traditional cigarettes in patients with schizophrenia spectrum disorders. About the meaning that cigarettes can assume for the patient, recurring themes emerge: they are considered as a way to manage nervousness and tension or as a means to contrast daily monotony and boredom or repeat usual gestures and habits.

## 1. Introduction

Schizophrenia is a chronic mental disorder, considered one of the first causes of mental disability worldwide [1]. Onset usually occurs between late adolescence and early adulthood. Symptoms persist for the whole life cycle; for this reason, it is one of the most serious and disabling psychological pathologies. The reality of the patient suffering from schizophrenia is altered in many aspects: the perceptive area is affected by hallucinations and the thought by delusions. This condition pours on the emotional and social front, where the patient manifests flattening and reduction of emotional expression and abnormal social behavior. Regarding the etiopathogenesis of schizophrenia, today’s science tends to support a polygenic hypothesis, so mutations of different genes, often rare, appear as determinants. In a study published by Purcell, genetic sequences of 2500 subjects in the schizophrenic population were compared with subjects in the normal population. Thanks to this comparison between several alterations of dozens of genes, it has been seen that the genes coding for the proteins involved in the formation of ion channels and those responsible for synaptic transmission are those that have a more important role [2]. Today, researchers have found that the most important genes playing a significant part in the onset of schizophrenia are COMT, NRG1, DISC1, and SETD1A, while the most relevant environmental risk factors affecting genetic predisposition are pre and peri-natal complications, substance abuse and stressful events [3].

The abuse of drugs, alcohol, and nicotine is particularly widespread among patients suffering from schizophrenia spectrum disorders, more than among the general population. As a result of substance use, there may be a worsening of psychotic symptoms, non-adherence to treatment, interaction with prescribed agents as well as an increase in violent behavior and suicide with premature mortality. These critical issues have been reported by several studies, highlighting the importance of finding solutions that can help patients improve their quality of life and that of the people around them, such as family and friends [4]. Focusing on smoking addiction, some studies have demonstrated that this problem affects between 60–90% of people with schizophrenia, compared to 15–24% of the non-psychiatric adult population [5]. From the literature, it emerges that genetic factors contribute more than all other factors, such as environmental or stress-related ones, to the comorbidity between smoking and schizophrenia, although it is necessary to underline the importance of further research to better understand the complex relationship between these two disorders [6]. The association between smoking and mental health conditions becomes stronger depending on the severity of the disease and seems to vary depending on the actual health conditions, so a person who has no physical or psychological health problems will be less likely to take substances [7].

People with schizophrenia spectrum disorder have an increased mortality risk if compared to the non-psychiatric adult population, and the risk is even higher in people with a double diagnosis of Schizophrenia and Substance use disorder [8]. In a large-scale follow-up study, Callaghan and colleagues also found that tobacco-related conditions included about 53 of total deaths in the schizophrenic cohort, 48% in the bipolar cohort, and 50% in the depressive cohort [9]. This is further confirmed by Carney’s studies which showed that people with schizophrenia spectrum disorders have a significantly higher prevalence of cancer, respiratory disease, and cardiovascular disease than the general population [10]. Regarding the effects that smoking can have on pathology among smokers with schizophrenia spectrum disorders, the association with depressive symptoms, increased hospitalization, stress, poor treatment results, low quality of life and improved psychotic symptoms has been noted [11]. The treatment of smoking addiction in people suffering from a psychiatric disorder is more complex than it is in non-psychiatric patients, and it is often neglected for other seemingly more important problems. Scientific literature has shown that the most effective treatments for these patients include nicotine replacement therapy (NTR), Bupropion, and behavioral interventions (CBT) [12]. In addition to these treatments, electronic cigarettes can be considered a low-risk substitute for traditional cigarettes, as has been demonstrated recently. Researchers have shown that the use of electronic cigarettes in patients with schizophrenia caused a 50% reduction in daily tobacco cigarette consumption in 50% of participants (from 30–15 cigs/day), whereas 14.3% of participants quit completely by week 52 [13]. Treatments are very important not only to improve the quality of life of these patients but also to help their family and friends who experience a condition of psychological discomfort due to cigarette abuse by a family member. A study by Budiono indicates that family psychoeducation on the nature of schizophrenia is effective in positively changing family reactions and indirectly increasing patients’ adherence to prescribed treatment [14]. Family members help and support the patient and assist in ensuring drug adherence. In our previous research, we analyzed the point of view of healthcare personnel dealing with schizophrenic patients who smoke and live in assisted therapeutic communities. Our results showed that all specialists agree on e-cigarettes as good alternatives to cigarettes. The positive view on electronic cigarettes is reinforced by other factors, such as that they don’t release bad smells and don’t stain or burn clothes and linens [15]. The current research has focused on family and friends of patients diagnosed with schizophrenia to better understand the impact that cigarette smoking has on the thoughts and emotions that relatives experience every day. We have studied the views of family members on traditional cigarettes, but also their consideration of e-cigarettes and the impact those could have in helping patients to decrease or stop smoking. We also investigated if participants had ever considered it appropriate to intervene in an attempt to reduce the number of cigarettes smoked by their family members and examined the risk perception of smoking on health.

## 2. Materials and Methods

This qualitative research was carried out between November 2021 and June 2022. It involved family and friends of patients with schizophrenia spectrum disorders living in two assisted therapeutic communities. The sample consisted of 30 participants, including 17 women and 13 men, who were selected by the non-probability sampling technique widely used in qualitative research, so participants are selected based on the basis of the research questions: be friends or relatives of adults who smoke with schizophrenia [16]. The methodological orientation was the analysis of the contents. No subjects refused to participate. Some participants were recruited by telephone, while another part was recruited from within the same communities. This is because some of them had returned to their homes or had been hospitalized for health problems. The inclusion criterion consisted in having a kinship/friendship/sentimental bond, such as to guarantee a thorough knowledge of the patients, their pathology and their level of dependence.

The participants were informed about the objectives of this qualitative analysis:The perception of family and friends of patients suffering from schizophrenia on smoking and its impact on physical and mental health;Their contribution to promoting or containing smoking addiction of the family member;Their perception of electronic cigarettes as an alternative to traditional cigarettes to reduce or eliminate dependency.

Based on the research questions, we developed more detailed questions that were asked through a semi-structured interview (Table 1). The interviews lasted about 10 min, depending on the subject’s answers.

Brief information was collected about the participants (Table 2): sex, degree of kinship or friendship. Then, they were informed about the purpose of our research, and they were asked for permission to record the interview, ensuring that the audio recording was used for the sole purpose of transcription and analysis. The semi-structured interviews were conducted individually. All interviews were recorded and subsequently transcribed. The length of the interviews ranged from 5 to 10 min, depending on the responses of participants. Audio recording has been very important for data collection purposes; in fact, during the same interview, it was preferred not to take notes of the answers in order to keep the interaction as natural and smooth as possible and be able to adapt the subsequent question to the previous answer of the participant.

We used the method of Thematic Analysis to identify, analyze, and report the issues within the data, as well as to provide a rich and detailed but complex data report [17].

The Thematic Analysis model consists of six phases:Familiarization: requires the researcher to develop a thorough knowledge of the data, familiarize himself with the material collected, reading the transcripts several times, with the aim of going beyond the most obvious meanings;Generation of codes: means identifying elements of interest, summarizing them and identifying them by codes, understood as words or short sentences. Braun and Clarke argue at this stage the importance of looking beyond the overt meaning of the participants’ answers to questions to also address the implicit and latent meaning;First generation of themes: Topics are being identified based on how the codes are combined with each other, so we proceed by identifying broader themes that group together different codes. This creates a list of themes with the corresponding codes;Revision of themes: This phase is very important to verify that the themes have remained consistent and data-related, plus it serves to rework the initial themes and decide whether to expand them into more units, delete them, group multiple themes into a larger unit or keep them as they are;Definition and naming of themes: each theme, which is identified in its essential characteristics, is assigned a label to name it;Production of the Report: A final report is produced at this stage, focusing more on the themes that answer the research questions, developing them and including extracts from the interviews to make the meaning of the themes and the report explicit, which must have been maintained between these and the data initially collected (Table 3).

## 3. Results

Most participants (25 out of 30) have a negative view of smoking, whether they are people who smoke daily or not. The same believe that the impact of cigarette smoking traditions is strongly negative on physical health, being the cause of several respiratory and cardiovascular disorders in patients. Some of them (18 out of 25) identify a correlation between the state of mental health of the patient and cigarette smoking, reporting that the patient has a greater need to smoke in moments of discomfort or nervousness. Participants who do not believe that smoking has a negative impact are all people who smoke daily (5 out of 30). They tend to underestimate the problem of the addiction, not attributing to smoking the responsibility of any type of symptom of the patients. Instead, as emerges from the interviews, they are more concerned about other types of problems, such as the unpredictable behavior of the patient, so they do not think the smoking addiction is a priority to be addressed. No participant reports a positive view of cigarette addiction. According to family and friends, recurrent themes emerge of cigarettes as a way to manage nervousness and tension or as a means to contrast daily monotony and boredom or repeating gestures and habits.

Twenty participants out of the total believe they have provided help to the patient, trying to make them stop smoking, intervening more or less directly (Theme 2). Some say they have recommended quitting, and others have intervened by interfering with the purchase of cigarettes or by establishing with the patient a maximum number of cigarettes a day, or even proposing alternatives to traditional cigarettes.

Ten patients did not intervene for different reasons: being people who smoke themselves, they did not think they could help the patient; they did not consider smoking a major problem; they did not believe that the patient could ever stop smoking. Only three participants believe that the patient can quit smoking, 20 argue that the patient can at least reduce, and seven report the impossibility that the patient would stop or reduce the habit of smoking. 

All participants reported knowing about e-cigarettes (Theme 3). Twenty of them think that e-cigarettes are less harmful than traditional cigarettes, seven felt unable to make a judgment because they said they did not have sufficiently thorough knowledge, while 3 participants considered e-cigarettes more dangerous than traditional cigarettes. Regarding the possibility of using e-cigarettes in an attempt to help the patient to quit smoking, 13 participants declared that they could be useful, while the remaining part believes they are not.

The analysis of the three themes was carried out after transcribing the interviews. Most participants, including adult who smoke daily, believe they have made a positive contribution to reducing patient dependency with either directional or authoritarian interventions. A small portion of the sample has never intervened for several reasons; they believe that any attempt is useless because they do not consider smoking such a problem as to require intervention or because, being people who smoke themselves, they don’t feel able to help another person with the same problem. Most participants believe that with the help of specialists, patients can quit smoking or at least reduce it, while a relatively small part of the sample thinks that this cannot happen.

## 4. Discussion

The influence of family or other significant people seems to be very important in two main aspects of dependence: first of all, in the development of the addiction and secondarily in the treatments and the interventions to contrast it.

About the first point, many theories and studies have shown how parenting has a strong impact in predicting exposure to deviant companions during adolescence and so in increasing the risk of developing deviant behaviors, such as smoking addiction and drugs dependencies [18]. Besides, several pieces of evidence testify to the role of the attachment relationship between members of the family and children. Both in the development of psychopathology and in the growth of dependence, systemic theory, in fact, has made several contributions to the understanding of addiction phenomena by shifting the focus of analysis from the subject to the family context, and in particular to the styles of attachment in intra-family relation [19]. Many studies have shown that safe attachment is an important protective factor against the development of mental disorders, while unsafe styles are associated with their development and consumption behaviors associated with addiction to tobacco and other substances [20].

A large body of research has provided evidence about the link between parental smoking and smoking in adolescents. Cognitive theories have tried to explain the reason behind this phenomenon, elaborating a research line that describes an explicit transmission path involving conscious elaboration and explicit decision-making processes. Theories that focus on health behavior, such as Social Cognitive Theory, assume that the decision to engage in substance use is based on a rational assessment of the positive and negative consequences. According to these theories, explicit cognitions are important predictors of the decision to engage in certain behaviors [21]. The observation of a certain behavior by a model figure forms cognition and, therefore, leads to the adoption and imitation of the same behavior [22]. Recent literature describes a second possible pathway that also considers the role of implicit smoking-related cognitions: implicit transmission. It involves the formation of cognitive processes that are more automatic and less readily accessible by introspection or self-report [23]. The achievement of the patient’s desired outcome can be considered a phenomenon widely studied in social psychology on self-fulfilling prophecy [24]. In this perspective, the beliefs that everyone has about themselves are strongly influenced by the expectations and beliefs of others. So applying this hypothesis to our research context, the beliefs of others, especially people considered as meaningful to the patient, like friends or family, can affect the real possibility of quitting smoking for the patient.

Regarding the second point, concerning the role of the family in the treatment of dependence, it is known that the behavior of one member of the family influences the behavior of other members.

So, according to the theory of the role model, individuals compare themselves with people that are significant to them, and these people are more likely to influence their behavior. For this reason, it is appropriate to think that the influence of family and other significant people may be important in the treatment of tobacco addiction because they are seen as models to imitate and to listen to. So, in an intervention program to limit cigarette consumption in schizophrenic patients, it can be very fruitful to include families and work with them to help the patients to achieve the goal of reducing or quitting smoking. The main objectives should be to make families conscious of their active role and their influence in this process and work on their perception of the risks associated with tobacco addiction.

It has also been demonstrated that a positive and supportive environment in the planning of the intervention can be useful to help the patient not to feel alone and helpless. Therefore, also thanks to the support of their family and friends, people with schizophrenia can proceed toward recovery and improve the quality of their leaves, which is the main goal of treating both schizophrenia and addiction [25].

Depending on the results of the current research, it might be interesting to conduct a large study focused on the perception of low-risk products also among doctors and health professionals. The lack of accurate communication from scientists may contribute to the confusion about the health risks deriving from e-cigarettes and their contribution to helping to quit the smoking habit. Electronic cigarettes could be, instead, a valuable ally in the fight against smoking addiction, especially in patients suffering from schizophrenia. Promoting their use might avoid authoritarian methods that would end up exacerbating the paranoid characteristics of the disorder. The use of e-cigarettes could give the patients the possibility to maintain a habit, which in many cases is an important part of their lives, while helping them to reach the first goal, which is risk reduction.

Eliminating common misconceptions about e-cigarettes is an important step that health professionals should consider in order to effectively educate the general population by providing the correct information about the risks of vaping compared to the risk of traditional tobacco. Some principles and solutions that should guide policymakers towards a more reasonable regulation of e-cigarettes include the concept of harm reduction, the possibility of de-normalization of smoking, the availability of low-risk products, the importance of proportionate taxation and of consumer perspectives, and the value of a reassessment of the role of non-tobacco flavors [26]. It is necessary to spread correct information about devices without combustion because, as evidenced by our research, behaviors, as well as the beliefs of patients, are often influenced by those of people significant to them. Several studies have shown that the harm perception of electronic cigarettes can deter people who smoke currently from starting or continuing the use of electronic cigarettes [27]. This perception can also discourage a complete switch from cigarettes to electronic cigarettes among adults who smoke. E-cigarettes share many similarities with smoking in the behavioral aspect of their use [28]. As confirmed by the data from clinical trials [28], users report the necessity of using an e-cigarette for a long period to reduce cigarette consumption or quit smoking or relieve tobacco withdrawal symptoms [29], with much-reduced health risks [30]. Moreover, the fact that they can be used in many areas where smoking is prohibited, that they have a very competitive price, and are perceived as a much less harmful alternative to tobacco smoking [31], increases their popularity. As Farsalinos demonstrated in his study, over 48 million people in the EU have ever used e-cigarettes, and 7.5 million currently use them [32]. Between 2012 and 2017, the proportion of those currently smoking tobacco in the EU who had ever tried to quit tobacco smoking using e-cigarettes increased from 7.1% to 15.6%, respectively [33]. The percentages of subjects who vape more than others are young, as shown by the many studies conducted in different parts of Western countries.

Clinicians and health policy experts who fight against the risks related to smoking cannot consider this phenomenon as an opportunity to improve people’s health. Increasing evidence confirms that harm reduction through the transition from traditional cigarettes to electronic products could represent an important stage in the path of smoking cessation [34], so it is equally good to promote a total cessation of smoking even with the abandonment of hard reductions tools, which although it is well known to represent an improvement from the point of view of danger and harmfulness about traditional tobacco, they must only represent a stage in the path of cessation and not a final point of arrival. However, the literature suggests with increasing evidence that the health risks are significantly lower than those of traditional smoking among adults who cannot or do not want to quit [35]. Innovation in the electronic vapor category is likely to further reduce health risks in favor of maximizing the health benefits of consumers switching from traditional to electronic cigarettes [36]. Instead, for people who wish to stop vaping, we must provide cessation programs that are effective, accessible, and well-publicized [37]. In the context of medical care, it is necessary to be clear about the benefits and options available for tobacco cessation, and test-based methods for this purpose should be easily accessible at the national level and free of charge.

## 5. Conclusions

This qualitative study compared the vision of 30 participants who had a relationship of kinship, friendship or sentimental nature on traditional cigarette smoking and low-risk products, including electronic cigarettes, in patients suffering from Schizophrenia spectrum disorders. We have also investigated their beliefs about tobacco abuse and new treatments, including the use of electronic cigarettes. The results of this study show that the view of most participants on smoking is negative (83.3%, 25 out of 30), but despite this, not everyone considers the treatment of tobacco in patients with the above disorders to be of primary importance (33.3%, 10 out of 30; 66.6%, 20 out of 30). Finally, low-risk products and, in particular, electronic cigarettes are viewed by a good number of participants as a useful alternative to traditional cigarettes in the fight against smoking in patients suffering from Schizophrenia Spectrum disorders.

Our research is certainly not without flaws or limitations. The small number of the sample does not allow us to generalize the data, and therefore, we do not have the certainty that what has been found is valid for the entire population that the sample represents. Another potential limitation could be that we are personally involved in data collection, as this increases the possibility of unconsciously influencing responses. The risk is to administer the questions, suggesting answers rather than letting them reflect the real perceptions of the participants. To avoid this, we tried to ask the questions in a similar way, keeping the same form, that is, the one reported in the interview. Future research will help us to fill these gaps and explore the phenomenon under study by using other types of research tools.

## Figures and Tables

**Table 1 healthcare-11-00644-t001:** Questions for research objectives.

First Objective
1. What do you think about smoking?
2. How dangerous do you consider this addiction to health?
3. Are you a person who smokes, or have you ever smoked?
4. How do you assess the physical and mental health of your family member/friend?
5. Do you think his/her health is related to your addiction to smoking?
6. Did you ever notice if the times when the addiction was more or less accentuated coincided with periods when the patient’s health was different?
7. Do you think there are times when the patient needs to smoke more?
8. Do you think that cigarette use can relieve or accentuate some symptoms of the patient’s mental illness?
9. What do you think cigarette represents for the patient?
**Second Objective**
1. If you are a person who smokes daily, do you think this will affect your familiar’s addiction? If so, how?
2. Have you ever tried to help him/her reduce or stop smoking? How?
3. Do you think he/she can actually do that?
**Third Objective**
1. Do you know about electronic cigarettes?
2. Do you think it is less, equally or more dangerous than traditional cigarettes?
3. Do you think it is a good alternative to help patients quit smoking?

**Table 2 healthcare-11-00644-t002:** Participants’ characteristics.

Gender	Degree of Kinship	Degree of Friendship
Male *n*(%)		
13 (43.34%)	Fathers 2 (6.65%)	Friends 6 (20%)
	Brothers 2 (6.65%)	Companions 3 (10%)
	Husbands 0 (0%)	
Female *n* (%)		
17 (56.66%)	Mothers 5 (16.65%)	Friends 0 (0%)
	Wives 2 (6.65%)	Companions 5 (16.65%)
	Sisters 5 (16.65%)	

Sample Size (*n*) = 30; Standard deviation (s.d.) = 1.669; Signify (x) = 3.75; Variance (s^2^) = 2.79; Coefficient variance = 0.4451; Average standard error (SE) = 0.59.

**Table 3 healthcare-11-00644-t003:** Themes and codes.

Research Questions	Themes	Codes
The perception of family and friends about smoking and its impact on the patient’s health	Positive, negative, or neutral perceptions of traditional cigarettes	- Participants who smoke daily/Participants who do not smoke - Assessment of the risk of smoking to physical and mental health - Relationship between difficult times and increased prevalence of cigarette addiction - Meaning which patient attributes to the cigarette
The contribution of family and friends in promoting or containing patient’s tobacco dependence	Positive, negative, or neutral impact on dependence	- Presence/absence of support for smoking cessation - Belief that patients can reduce or quit smoking
Family and friends‘ perception of e-cigarettes and their contribution to reducing addiction	Perception of the electronic cigarette	- Knowledge about electronic cigarette - Impact of e-cigarettes on health - Assessment of electronic cigarettes as a valid means of containing addiction or not

## Data Availability

Not applicable.

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
