# Peer review of "Psychological Experience of Smoking Addiction in Family and Friends of Schizophrenic Adults Who Smoke Daily: A Qualitative Study"

_healthcare, 2023, doi:10.3390/healthcare11050644_

Round 1

Reviewer 1 Report

The article is good written, it is important topic and I like the manuscript .

My suggestion is to add and prepare a table that include the important novel points of this article. So the reader will see a summary of this article and the new findings organised. In the end my suggestion is to add the finding of this article as points if possible.    

Reviewer 2 Report

Thank you for giving me the opportunity to review the manuscript “Psychological experience on smoking addiction in family and friends of smokers with Schizophrenia Spectrum Disorders: a qualitative research”. The methodological section of the manuscript has many inadequacies. My suggestions are here to consider.

Abstract: Why did the author mention the term ‘Survey’? this is a semi-structured interview (qualitative study). No information on data collection and analysis tools and study implication

Introduction

Many claims have no citations, for example: “The abuse of drugs………………………..the general population” This must be supported by the reference.

Lack of coherency. For example: “In a study published by Purcell………………Thanks to this comparison between ……………. important role”. This sentence especially “Thanks to this comparison between” sentence doesn’t reflect academic writing skill.

Method

Which author/s conducted the interview or focus group? What were the researcher’s credentials?

What methodological orientation was stated to underpin the study? grounded theory/ethnography/ phenomenology/content analysis

Why Some participants were recruited by telephone and another recruited from within the same communities?

How many people refused to participate or dropped out? Reasons?

What was the duration of the interviews?

Was data saturation discussed?

How many data coders coded the data?

What software, if applicable, was used to manage the data?

Discussion: Study limitations, strengths, and clinical implications are missing

Grammar and spelling checks need to be properly revised.

Round 2

Reviewer 2 Report

Thank you for giving me the opportunity to review the revised manuscript of “Psychological experience on smoking addiction in family and friends of smokers with Schizophrenia Spectrum Disorders: a qualitative research”. The thematic analysis method should report in the methodology section. 

Author Response

Dear reviewer, as you requested the thematic analysis procedure has been moved to the methodological section.

Kind regards. 

Reviewer 3 Report

Thank you for the chance to review the revised manuscript. The manuscript has been sufficiently improved. 

I only have one minor revision suggestion. For Table 2, author used N (%) for male and female, which is very good. But for the data, for example, Male it shows (13) 43.34%, Fathers, it shows (2) 6.65%, Friends (6) 20%. I think it should be Male 13 (43.34%), Fathers 2 (6.65%) and Friends 6 (20%). 

Author Response

Dear Reviewer, Please excuse us for the error. We fixed to change the data in Table 2. 

Kind regards.